# Feynman Rules, Ward Identities and Loop Corrections in Very Special Relativity Standard Model †

**Jorge Alfaro** 

Instituto de Física, Pontificia Universidad Católica de Chile, Santiago 7820436, Chile; jalfaro@fis.puc.cl;
Tel.: +56-22-354-4483
† This paper is based on the talk at the 7th International Conference on New Frontiers in Physics (ICNFP 2018), Crete, Greece, 4–12 July 2018.

**Abstract:** In this paper, we want to study one loop corrections in Very Special Relativity Standard Model(VSRSM). In order to satisfy the Ward identities and the $Sim(2)$ symmetry of the model, we have to specify the Feynman rules, including the infrared regulator. To do this, we adapt the Mandelstam–Leibbrandt (ML) prescription to incorporate external momentum-dependent null vectors. As an example, we use the new $Sim(2)$ invariant dimensional regularization to compute one loop corrections to the effective action in the subsector of the VSRSM that describe the interaction of photons with charged leptons. New stringent bounds for the masses of $\nu_e$ and $\nu_\mu$ are obtained.

**Keywords:** Feynman rules; very special relativity; Ward identities

## 1. Introduction

The Weinberg–Salam model of weak and electromagnetic interactions (SM) is being verified at the LHC: the discovery of the Higgs particle ratified the mechanism of spontaneous symmetry breaking (SSB). However, the mark of new particles or new interactions do not appear at the LHC at present [1]. Nevertheless, some anomalies have been found in the lepton flavor universalities of the semileptonic B decays, which could point to new physics [2–9].

However, we know that the SM is not adequate in the neutrino sector. In the SM, neutrinos are massless, whereas in Nature, at least some of the species of neutrinos must be massive because they exhibit neutrino oscillations [10].

To describe massive neutrinos in a Lorentz-invariant quantum field theory (QFT), a new kind of particles must be postulated as in the seesaw mechanism [11].

If we want to keep the particles and symmetries of the SM and give masses to the neutrinos, one possibility is very special relativity (VSR) [12]. VSR postulates that the true symmetry of Nature is not the Lorentz group (six parameters), but a subgroup of it, $Sim(2)$ (four parameters). Using this subgroup, none of the classical tests of special relativity (SR) are affected, but it is possible to provide a non-local mass term for the neutrino [13].

Recently, we have used VSR to build the VSRSM [14]. We have used the same particles and symmetries as the Weinberg–Salam model, but masses and neutrino oscillations are explained by a non-local Lorentz-violating and $Sim(2)$-invariant contribution. New predictions are obtained such as the process $\mu- > e + \gamma$, which is forbidden in the SM.

In this paper, we want to consider loop corrections in the VSRSM. To do that, we need to introduce appropriate Feynman rules, including an external momentum-dependent null vector for the infrared regulator in the Mandelstam–Leibbrandt prescription [15–17]. In this note, we calculate one loop corrections in very special relativity quantum electrodynamics (VSRQED), which are obtained restricting the VSRSM to the processes involving photons and charged leptons alone. The charged

lepton and the neutrino of the same family share a common VSR mass (they are a $SU(2)_L$ doublet). Then, we are able to get information about the mass of the neutrino calculating transitions in VSRQED.

## 2. The Model

At present, we restrict ourselves to the electron family. $m$ is the VSR mass of both the electron and neutrino, producing the second term of (1). After spontaneous symmetry breaking (SSB), the electron acquires a mass term $M = \frac{G_e v}{\sqrt{2}}$, where $G_e$ is the electron Yukawa coupling and $v$ is the Vacuum Expectation Value(VEV)of the Higgs, i.e., the third term of (1). Please see Equation (52) of [14]. The neutrino mass is not affected by SSB: $M_{\nu_e} = m$.

Restricting the VSRSM after SSB to the interactions between the photon and electron alone, we get the VSRQED action. $\psi$ is the electron field. $A_\mu$ is the photon field. We use the Feynman gauge.

$$\mathcal{L} = \bar{\psi}\left(i\left(\not{D} + \frac{1}{2}\,\not{n}m^2(n\cdot D)^{-1}\right) - M\right)\psi - \frac{1}{4}F_{\mu\nu}F^{\mu\nu} - \frac{(\partial_\mu A_\mu)^2}{2} \tag{1}$$

In (1), $n_\mu$ is a fixed null vector, i.e., $n.n = 0$, $D_\mu = \partial_\mu - ieA_\mu$, $F_{\mu\nu} = \partial_\mu A_\nu - \partial_\nu A_\mu$.

We see that the electron mass, determined by the pole of the propagator, is $M_e = \sqrt{M^2 + m^2}$. In fact, Dirac's equation in momentum space is:

$$\left(\not{p} - M - \frac{m^2}{2}\frac{\not{n}}{n\cdot p}\right)u(p) = 0 \tag{2}$$

Multiplying by $\left(\not{p} + M - \frac{m^2}{2}\frac{\not{n}}{n\cdot p}\right)$, we get:

$$\begin{aligned}\left((\not{p} - \frac{m^2}{2}\frac{\not{n}}{n\cdot p})^2 - M^2\right)u(p) &= 0 \\ \left(p^2 - m^2 - M^2\right)u(p) &= 0\end{aligned} \tag{3}$$

We have used that $\not{n}.\not{n} = n.n = 0$.

To draw the Feynman graphs we used [18]. In the following sections, we have extensively used the program FORM [19]. Due to the non-local term $(n.D)^{-1}$, there is an infinite number of photon-electron vertices corresponding to the expansion:

$$\begin{aligned}(n.D)^{-1} &= (n.\partial - ien.A)^{-1} = (1 - ie(n.\partial)^{-1}(n.A))^{-1}(n.\partial)^{-1} = \\ &(1 + ie(n.\partial)^{-1}(n.A) + (ie)^2(n.\partial)^{-1}(n.A)(n.\partial)^{-1}(n.A) + \ldots)(n.\partial)^{-1}\end{aligned} \tag{4}$$

Inserting (4) into the action (1), we can derive the Feynman rules.

In Figure 1, we write the Feynman rules needed to compute one loop one-particle irreducible diagrams with at most two external photon legs.

**Figure 1.** Feynman rules for one loop computations: electron and photon propagators, $A_\mu ee$ and $A_\mu A_\nu ee$ vertex.

We notice that the fermion propagator, as well as the vertices contain $n.p_i$ in the denominators. $p_i$ are momenta.

These factors will produce infrared divergences in loop integrals. To regularize these integrals, we will use the Mandelstam–Leibbrandt prescription (ML) [15–17].

## 3. Mandelstam–Leibbrandt Prescription

As an example, let us compute the following simple integral:

$$A_\mu = \int dp \frac{f(p^2) p_\mu}{n.p}$$

where $f$ is an arbitrary function. $dp$ is the integration measure in $d$-dimensional space, and $n_\mu$ is a fixed null vector ($n.n = 0$). This integral is infrared divergent when $n.p = 0$.

To regulate the infrared divergence, we use the ML prescription:

$$\frac{1}{n.p} = \lim_{\varepsilon \to 0} \frac{p.\bar{n}}{n.p \, p.\bar{n} + i\varepsilon} \tag{5}$$

This prescription has very nice features: It is invariant under the shift of the integration variable, preserves naive power counting of divergent integrals, combines well with dimensional regularization [17], and is the unique prescription that can be derived from canonical quantization [20].

To compute $A_\mu$, we have to know the specific form of $f$, provide a specific form of $n_\mu$ and $\bar{n}_\mu$, and evaluate the residues of all poles of $\frac{f(p^2)}{n.p}$ in the $p_0$ complex plane, a rather formidable task for an arbitrary $f$.

Instead, we want to point out the following symmetry [17]:

$$n_\mu \to \lambda n_\mu, \bar{n}_\mu \to \lambda^{-1} \bar{n}_\mu, \lambda \neq 0, \lambda \varepsilon R \tag{6}$$

It preserves the definitions of $n_\mu$ and $\bar{n}_\mu$:

$$0 = n.n \to \lambda^2 n.n = 0$$
$$0 = \bar{n}.\bar{n} \to \lambda^{-2} \bar{n}.\bar{n} = 0$$
$$1 = n.\bar{n} \to n.\bar{n} = 1$$

We see from (5) that:

$$\frac{1}{n.p} \to \frac{1}{n.p} \lambda^{-1}$$

Now, we compute $A_\mu$, based on its symmetries. It is a Lorentz vector, which scales under (6) as $\lambda^{-1}$. The only Lorentz vectors we have available in this case are $n_\mu$ and $\bar{n}_\mu$. However, (6) forbids $n_\mu$. That is:

$$A_\mu = a\bar{n}_\mu$$

Multiply by $n_\mu$ to find $A.n = a$. Thus, $a = \int dp f(p^2)$. Finally:

$$\int dp \frac{f(p^2) p_\mu}{n.p} = \bar{n}_\mu \int dp f(p^2)$$

We consider now a more general integral. We will see here that the regularity of the answer will determine it uniquely.

Consider:

$$A = \int dp \frac{F(p^2, p.q)}{n.p} = \bar{n}.q f(q^2, n.q \bar{n}.q) \tag{7}$$

$q_\mu$ is an external momentum, a Lorentz vector. $F$ is an arbitrary function. The last relation follows from (6), for a certain $f$ we will find in the following.

We get:

$$\frac{\partial A}{\partial q^\mu} = \int dp \frac{F_{,u} p_\mu}{n.p} =$$

$$\bar{n}_\mu f(x,y) + 2\bar{n}.q q_\mu \frac{\partial}{\partial x} f(x,y) + [(\bar{n}.q)^2 n_\mu + n.q\bar{n}.q\bar{n}_\mu] \frac{\partial}{\partial y} f(x,y)$$

We define $u = p.q$, $x = q^2$, $y = n.q\bar{n}.q$. $()_{,u}$ means the derivative with respect to $u$. Thus:

$$\frac{\partial A}{\partial q^\mu} n_\mu = \int dp F_{,u} = \quad g(x) =$$

$$f(x,y) + 2y \frac{\partial}{\partial x} f(x,y) + y \frac{\partial}{\partial y} f(x,y) \tag{8}$$

Assuming that the solution and its partial derivatives are finite in the neighborhood of $y = 0$, it follows from the equation that $f(x,0) = g(x)$. That is, the partial differential equation has a unique regular solution.

Now, we apply this result to compute integrals that appear in gauge theory loops:

$$\int dp \frac{1}{[p^2 + 2p \cdot q - m^2]^a} \frac{1}{(n \cdot p)} = (\bar{n} \cdot q) f(x,y)$$

In this case:

$$g(x) = -2a \int dp \frac{1}{[p^2 - x - m^2]^{a+1}}$$

The unique regular solution of (8) is:

$$f(x,y) = -\frac{1}{y} \left\{ \int dp[p^2 - x - m^2]^{-a} - \int dp[p^2 - x + 2y - m^2]^{-a} \right\}$$

We can check that $f(x,0) = -2a \int dp[p^2 - x - m^2]^{-a-1} = g(x)$.

Using the same technique and dimensional regularization, we obtain, for arbitrary complex $a, b, \omega$:

$$\int dp \frac{1}{[p^2 + 2p.q - m^2]^a} \frac{1}{(n.p)^b} =$$

$$(-1)^{a+b} i(\pi)^\omega (-2)^b \frac{\Gamma(a+b-\omega)}{\Gamma(a)\Gamma(b)} (\bar{n}.q)^b \int_0^1 dt\, t^{b-1} \frac{1}{(m^2 + q^2 - 2n.q\bar{n}.qt)^{a+b-\omega}}, \quad \omega = d/2 \tag{9}$$

Other integrals can be obtained deriving with respect to $q_\mu$:

$$\int dp \frac{p_\mu}{[p^2 + 2p.q - m^2]^{a+1}} \frac{1}{(n.p)^b} =$$

$$(-1)^{a+b} i(\pi)^\omega (-2)^{b-1} \frac{\Gamma(a+b-\omega)}{\Gamma(a+1)\Gamma(b)} (\bar{n}.q)^{b-1} b\bar{n}_\mu \int_0^1 dt\, t^{b-1} \frac{1}{(m^2 + x - 2yt)^{a+b-\omega}} +$$

$$(-1)^{a+b} i(\pi)^\omega (-2)^b \frac{\Gamma(a+b+1-\omega)}{\Gamma(a+1)\Gamma(b)} (\bar{n}.q)^b \int_0^1 dt\, t^{b-1} \frac{q_\mu - t(n.q\bar{n}_\mu + \bar{n}.qn_\mu)}{(m^2 + x - 2yt)^{a+b+1-\omega}} \tag{10}$$

ML introduces an extra null vector $\bar{n}$, such that $n.\bar{n} = 1$ Having a fixed global extra null vector $\bar{n}_\mu$ destroys the $Sim(2)$ invariance of the model. To avoid this problem, we have to trade $\bar{n}_\mu$ by a linear combination of $n_\mu$ and some momenta $P_\mu$, which are external to the loop, as we did in [21], i.e., $\bar{n}_\mu = an_\mu + bP_\mu$. From the conditions: $\bar{n}.\bar{n} = 0$, $\bar{n}.n = 1$, we get $\bar{n}_\mu(P) = -\frac{P^2}{2(n.P)^2} n_\mu + \frac{P_\mu}{n.P}$.

We will see below that it is possible to assign different $\bar{n}_\mu(P_i)$ inside a one-particle irreducible (1PI) Green function such that the Ward identity and the $Sim(2)$ symmetry are preserved. Here, $P_i$ are linear combinations of the external momenta.

## 4. Tree Graphs

We assume that each fermion and boson line carries a pair of vectors $(p, P)$. $p$ is the usual momentum, whereas $P$ is an associated momentum used to build $\bar{n}(P)$. External fermion and boson legs (not attached to a fermion loop) have $P = p$.

Both momenta $p$ and $P$ are conserved at each vertex.

A boson line connecting two points in the same fermion line has $P = 0$.

The three-vertex is now:

$$- ie \left\{ \gamma_\mu + \frac{1}{2} n_\mu m^2 \; \not{h}(n.(p+q))_{P_2}^{-1}(n.q)_{P_1} \right\} \tag{11}$$

$P_1(P_2)$ are the associated momentum corresponding to the incoming (outgoing) fermion leg arriving at the vertex.

*Current Conservation*

Define:

$$\bar{u}_2(p') \left\{ (\gamma_\mu)_{\beta\alpha} + \frac{1}{2} (\not{h})_{\beta\alpha} \, n_\mu m^2 \left[ \left( \frac{1}{n.p} \right)_1 \left( \frac{1}{n.p'} \right)_2 \right] \right\} u_1(p) = j_\mu \tag{12}$$

We find that the current is conserved:

$$(p' - p)_\mu j_\mu = \bar{u}_2(p') \left\{ \not{p}' - \not{p} + \frac{1}{2} (\not{h}) m^2 \left[ \left( \frac{1}{n.p} \right)_1 - \left( \frac{1}{n.p'} \right)_2 \right] \right\} u_1(p) = 0 \tag{13}$$

We have used that:

$$\bar{u}_2(p') \left[ \not{p}' - \frac{1}{2} (\not{h}) m^2 \left( \frac{1}{n.p'} \right)_2 - M \right] = 0 \tag{14}$$

$$\left[ \not{p} - \frac{1}{2} (\not{h}) m^2 \left( \frac{1}{n.p} \right)_1 - M \right] u_1(p) = 0 \tag{15}$$

and:

$$n.p \left( \frac{1}{n.p} \right)_{\bar{n}} = \lim_{\varepsilon \to 0} \frac{(n \cdot p)(p \cdot \bar{n})}{(n \cdot p)(p \cdot \bar{n}) + i\varepsilon} = 1, \forall \bar{n} \tag{16}$$

The last identity holds even if $p$ is integrated over.

That is, external legs can be defined using different $\bar{n}$ for each, and the gauge symmetry still will be preserved.

The assignation of $P_\mu$ for tree graphs respects momentum $(p, P)$ conservation in each vertex, for any number of insertions of photon lines. In tree graphs, all $P_\mu$ are known, in terms of the external momenta of photon and fermion lines.

## 5. Photon Self-Energy in VSRQED

In this section, we present the computation of the photon self-energy. It is easy to check that the Ward identity for any number of external photon lines attached to a fermion loop implies that all $\bar{n}$ in the loop are equal. Since the Ward identity implies that the 1PI graph must be symmetric under the interchange of all independent external photon momenta, we choose $\bar{n}_\mu(P) = -\frac{P^2}{2(n.P)^2} n_\mu + \frac{P_\mu}{n.P}$, with $P_\mu = \sum_i k_{i\mu}$, where $k_{i\mu}$ is a complete set of independent external photon momenta.

We consider now the graphs contributing to the photon self-energy:

Applying the $Sim(2)$-invariant regulator to the addition of the graphs of Figure 2, we get:

$$i\Pi_{\mu\nu} = A(\eta_{\mu\nu}q^2 - q_\mu q_\nu) + B\left(-q^2\frac{n_\mu n_\nu}{(n.q)^2} + \frac{n_\mu q_\nu + n_\nu q_\mu}{n.q} - \eta_{\mu\nu}\right) \tag{17}$$

with

$$A = (-ie)^2\frac{i}{(4\pi)^\omega}\int_0^1 dx\,\Gamma(2-\omega)\frac{8x(1-x)}{(M_e^2-(1-x)xq^2)^{2-\omega}}$$
$$B = -m^2 i\frac{e^2}{4\pi^2}\int_0^1\frac{dx}{(1-x)}\log\left[1-\frac{q^2(1-x)^2}{M_e^2-q^2(1-x)x}\right] \tag{18}$$

Here, $-e$ is the electron electric charge, $m$ the electron neutrino mass, and $M_e$ the electron mass. $q_\mu$ is the virtual photon momentum.

We first notice that $q^\mu\Pi_{\mu\nu} = 0$, as required by the $U(1)$ gauge invariance of the photon field. It is obtained by a straightforward application of the regularized integrals of [21]. Moreover, $B(q^2 = 0) = 0$; therefore, the photon remains massless. Furthermore, the photon wave function divergence is the same as in QED.

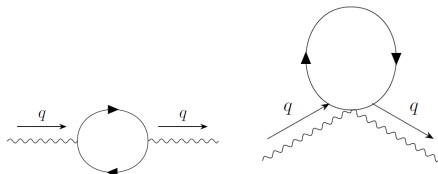

**Figure 2.** Vacuum polarization one loop graphs.

## 6. Electron Self-Energy in VSRQED

Here, we calculate the electron self-energy. Again, we have two graphs contributing to the two-proper vertex. See Figure 3.

$$-i\Sigma(q) = C\frac{\hbar}{n.q} + D\,\slashed{q} + E \tag{19}$$

with:

$$C = (-ie)^2 m^2\Big[\frac{i}{16\pi^2}\int_0^1 dx(1-x)^{-1}\ln\left(1+\frac{q^2(1-x)}{(M_e^2-q^2-i\varepsilon)}\right) +$$
$$2i(4\pi)^{-\omega}\int_0^1 dx\frac{\Gamma(2-\omega)}{[\mu^2 x - x(1-x)q^2 + M_e^2(1-x) - i\varepsilon]^{2-\omega}}\Big], \tag{20}$$
$$D = -2(-ie)^2(\omega-1)i(4\pi)^{-\omega}\int_0^1 dx\frac{\Gamma(2-\omega)x}{[\mu^2 x - x(1-x)q^2 + M_e^2(1-x) - i\varepsilon]^{2-\omega}}, \tag{21}$$
$$E = (-ie)^2 2\omega Mi(4\pi)^{-\omega}\int_0^1 dx\frac{\Gamma(2-\omega)}{[\mu^2 x - x(1-x)q^2 + M_e^2(1-x) - i\varepsilon]^{2-\omega}} \tag{22}$$

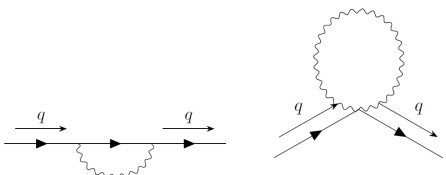

**Figure 3.** Electron self-energy one loop graphs. The second graph vanishes in the Feynman gauge.

## 7. Electron-Electron-Photon Proper Vertex) $\Gamma^\mu(p+q,p)$

In this section, we discuss the electron-electron-photon (EEP) vertex and verify the Ward–Takahashi identity. This is an important test of the gauge invariance of the regulator. The one loop contribution to $\Gamma^\mu(p' = p + q, p)$ consists of the addition of three graphs (Figure 4):

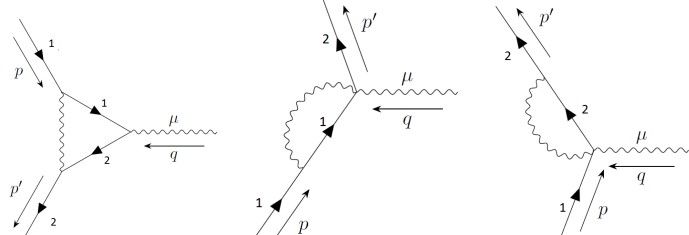

**Figure 4.** One loop contribution to the three-point proper vertex.

As a result of the shift symmetry, which is respected by the regulator, $\int dp f(p_\mu) = \int dp f(p_\mu + q_\mu)$ for arbitrary $q_\mu$, we can prove the Ward–Takahashi identity. This identity must hold for both the divergent and finite pieces of the Green functions:

$$- i q_\mu \Gamma^\mu(p+q,p) = S^{-1}(p+q) - S^{-1}(p) \tag{23}$$

Here, $S(p) = \frac{i}{\not p - M - \Sigma(p)}$ is the full electron propagator and $\Gamma^\mu(p+q,p)$ is the EEP vertex. The Ward–Takahashi identity is satisfied with the assignments shown in the graphs. The numbers represent different $\bar n$. According to the conservation rule for the extra momenta $P_\mu$, we must have: $P_{2\mu} = p'_\mu$, $P_{1\mu} = p_\mu$. As a simple check, below, we explicitly verified that the pole at $d = 4$ satisfies (23).

Pole contribution:

$$P\Sigma(q) = -(-ie)^2 \frac{1}{16\pi^2} \left\{ 2m^2 \frac{\not h}{n.q} - \not q + 4M \right\} \frac{1}{2-\omega} \tag{24}$$

$$P\Gamma^\mu(p+q,p) = -(-ie)^2 \frac{1}{16\pi^2} \frac{1}{2-\omega} \left( \gamma_\mu + 2m^2 \not h \frac{n_\mu}{n.p\; n.(p+q)} \right) \tag{25}$$

The divergent piece satisfies the Ward–Takahashi identity (23):

$$q_\mu P\Gamma^\mu(p+q,p) = P\Sigma(p) - P\Sigma(p+q) = -(-ie)^2 \frac{1}{16\pi^2} \frac{1}{2-\omega} \left\{ \not q - 2m^2 \not h \left( \frac{1}{n.p + n.q} - \frac{1}{n.p} \right) \right\}$$

*Form Factors*

The on-shell EEP vertex can be written as follows:

$$M_\mu(p,q) = \bar u(p+q) \left\{ G_2 \left[ -i\sigma_{\mu\nu} q_\nu\; \not h \right] + G_3\; \not h Q_\mu + F_3\; \not h \sigma_{\mu\nu} q_\nu\; \not h + \tilde\gamma_\mu F_1 + F_2 i \frac{\sigma_{\mu\nu}}{2M} q_\nu \right\} u(p) \tag{26}$$

where:

$$\tilde\gamma_\mu = \gamma_\mu + \frac{m^2}{2} \frac{\not h n_\mu}{n.p\;(n.p + n.q)}, \quad Q_\mu = q_\mu - q^2 \frac{n_\mu}{n.q} \tag{27}$$

$F_1, F_2.F_3, G_2, G_3$ are form factors (Lorentz scalar combinations of $n_\mu, p_\mu, q_\mu$). Under the $Sim(2)$ scaling $n_\mu \to \lambda n_\mu$, $F_1, F_2$ are invariants, $F_3 \to \lambda^{-2} F_3$, $G_3 \to \lambda^{-1} G_3$, $G_2 \to \lambda^{-1} G_2$.

In the non-relativistic (NR) limit, we get Table 1, keeping terms that are at most linear in $q_\mu$. The operator whose coefficient is the form factor reduces in this limit to the addition of the terms

appearing in the first column of Table 1, corresponding to the same form factor in the second column of Table 1.

**Table 1.** In the right column, we list the form factor. In the left column, we have the non-relativistic (NR) limit of the matrix element accompanying the form factor in (26). All form factors are evaluated at $q_\mu = 0$. Here, $A_0$ is the electric potential and $A_i$ is the vector potential. $\varphi_{s'}$ is a two- dimensional constant vector that corresponds to the NR limit of the Dirac spinors.

| NR Limit | Form Factor |
|---|---|
| $2M_e\varphi_s^\uparrow \varphi_s A_0$ | $F_1(0)$ |
| $\frac{3m^2}{4M^2} i\varepsilon_{ijk}\varphi_s^\uparrow \sigma^i \varphi_{s'} \hat{n}_j q_k A_0$ | $F_1(0)$ |
| $i\varepsilon_{ijk} q_j \varphi_s^\uparrow \sigma^k \varphi_{s'} A_i$ | $F_1(0)$ |
| $-2in_0 M\varepsilon_{ijk}\varphi_s^\uparrow \sigma^k \varphi_{s'} q_j A_i$ | $G_2(0)$ |
| $-i\varepsilon_{ijk} n_k \frac{m^2}{M}\varphi_s^\uparrow \hat{n}.\vec{\sigma}\varphi_{s'} q_j A_i$ | $G_2(0)$ |
| $i(2M\varepsilon_{ijk} n_k \varphi_s^\uparrow \sigma^j \varphi_{s'} + 2M_e i n_i \varphi_s^\uparrow \varphi_{s'})A_0 q_i$ | $G_2(0)$ |
| $2M_e n_0 \varphi_s^\uparrow \varphi_{s'} Q_\mu A^\mu$ | $G_3(0)$ |
| $(-4M_e\varepsilon_{ijk} n_k \varphi_s^\uparrow \vec{n}.\vec{\sigma}\varphi_{s'} + 4M_e n_0^2 \varepsilon_{ijk}\varphi_s^\uparrow \sigma^k \varphi_{s'})q_j A_i$ | $F_3(0)$ |
| $4M_e n_0 \varepsilon_{ijk} n_j \varphi_s^\uparrow \sigma^k \varphi_{s'} A_0 q_i$ | $F_3(0)$ |
| $i\varepsilon_{ijk}\varphi_s^\uparrow \sigma^k \varphi_{s'} A_i q_j$ | $F_2(0)$ |
| $-i\frac{m^2}{2M^2}\varepsilon_{ijk}\hat{n}_j \varphi_s^\uparrow \sigma^k \varphi_s A_0 q_j$ | $F_2(0)$ |

To show the power of the $Sim(2)$-invariant regularization prescription presented in [21], we will compute the one loop contribution to the (isotropic) anomalous magnetic moment of the electron. It is given by $F_2(0) - 2n_0 MG_2(0) - 4F_3(0)M_e n_0^2 i$ (see Rows 11, 5, and 9 of Table 1).

Evaluating the integrals according to the $Sim(2)$-invariantprescription to $o(m^2)$, we get [21]:

$$F_2 - 4F_3 M_e n_0^2 i - 2G_2 n_0 M = \frac{\alpha}{2\pi} \tag{28}$$

where $\alpha$ is the fine structure constant. Therefore, to this order, the QED result holds.

Notice that already at the tree level, the model predicts the existence of an anisotropic electric moment of the electron, corresponding to the second line of the list and an anisotropic magnetic moment of the electron, corresponding to the fourth row of the list, both of the order of $\frac{m^2}{M_e^2}$.

To compute the electric dipole moment, notice that the effective coupling of the electron to an external electric potential $A_0$ is given by the first row in Table 1, $2M_e\varphi_s^\uparrow \varphi_s A_0$. Therefore, to have the right coupling $eA_0$, we must redefine the NR spinors:

$$\sqrt{2M_e}\,\varphi_s = \chi_s \tag{29}$$

Then, the second row of Table 1 is,

$$\frac{1}{2M_e}\frac{3m^2}{2M^2}\vec{E}.(\hat{n} \times \vec{s}) \tag{30}$$

We have used that $E_k = -iq_k A_0$, $\chi_s^\uparrow \sigma^i \chi_{s'} = 2s^i$ and $n_0 = |\vec{n}|$, where $s^i$ is the electron spin. In the Born approximation, this corresponds to the coupling of the electron to an external electric field with the interaction energy $-\vec{p}.\vec{E}$. Then, the electric dipole moment is:

$$\vec{p} = -\frac{1}{2M_e}\frac{3m^2}{2M^2}(\hat{n} \times \vec{s}) \tag{31}$$

Therefore,

$$|\vec{p}| = \frac{3e}{4M_e}\frac{m^2}{M_e^2}|(\vec{s} \times \hat{n})| \leq \frac{3}{8}\lambda e\frac{m^2}{M_e^2} \tag{32}$$

where $\lambda = 2.4 \times 10^{-12} m$ is the Compton wavelength of the electron.

Using the best bound on the electric dipole moment of the electron [22], $|\vec{p}| < 8.7 \times 10^{-29}$ *eċm.*, we get:

$$\frac{m^2}{M_e^2} < 9.7 \times 10^{-19} \tag{33}$$

For the muon, $\lambda = 1.17 \times 10^{-14} m$. Using the best bound on the muon electric dipole moment [23], $|\vec{p}_\mu| < 1.8 \times 10^{-19}$ *eċm.*, we get:

$$\frac{m_\mu^2}{M_\mu^2} < 4 \times 10^{-7} \tag{34}$$

The $Sim(2)$-invariant regularization permits exploring the full quantum properties of VSR. They should be systematically tested, in particle physics models, as well as in quantum gravity models.

## 8. Conclusions

We applied the VSR formalism to the Weinberg–Salam model. This modification admits the generation of a neutrino mass without lepton number violation and without sterile neutrinos or other types of additional particles.

Now, we have non-local mass terms that violate Lorentz invariance.

The model is renormalizable, and the unitarity of the S matrix is preserved.

We studied the QED part of the VSR SM. Feynman rules were obtained. These rules incorporate the information about the infrared regulator compatible with the Ward identities.

We invented a $Sim(2)$-invariant dimensional regularization.

We computed the vacuum polarization graphs. They satisfy the Ward identity. New form factors in the vertex correction were computed. We obtained bounds on the mass of the neutrino from new form factors and the anomalous electric moment of the electron. $\frac{m^2}{M_e^2} < 9.7 \times 10^{-19}$.

**Funding:** The work of J.A. has been partially financed by Fondecyt 1150390 and Anillo ACT1417.

**Acknowledgments:** J.A. would like to thank the organizers of the "7th International Conference on New Frontiers in Physics (ICNFP 2018)" for a very beautiful and stimulating atmosphere; and Anna Pollmann, Nick Mavromatos, Dariusz Gora, and Alessandro Paoloni for very interesting conversations.

**Conflicts of Interest:** The author declares no conflict of interest.

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
