# Peer review of "Feynman Rules, Ward Identities and Loop Corrections in Very Special Relativity Standard Modelâ€"

_universe, doi:10.3390/universe5010016_

Round 1
Reviewer 1 Report
This is an interesting application of the ideas put forward in refs.[12,13], proposing a reduction of
the full Poincare symmetry within the Lorentz group to potentially mimick the effects of neutrino masses, to the computation of one-loop radiative effects in VSR QED. (The paper title is misleading since what is done in the paper is only VSR QED.)
I would suggest the following improvements before the paper is accepted.
1) Even though this is almost trivial, it may be beneficial to the less informed reader to show why the pole of the electron propagator subject to the nonlocal term in (1) is M_e=\sqrt{M^2+m^2} to lowest order in e.
2) It cannot hurt to be more explicit about the Mandelstam-Leibbrandt regularization since in Eqs.(5),(6),(7),(9) the number \omega(=2-d/2) appears without any introduction.
3) Fig.1 does not represent the entire vertex structure of the theorxy, rather a trunction of (n D)^{-1}
up to second order in A_\mu is presented. This should be mentioned and justified.
4) At several places a plural instead of a singular and other misprints appear:
differents -> different, Neanmoins -> Nevertheless, non -> none, preserve -> preserves, forms -> form, must holds -> must hold
5) In the third-last equation on p.6: \vec{s} is the spin vector (\hbar=1). How can this formula be obtained from the form-factor calculation? Please explain.
Upon satisfactory addressation of the above points the paper can be published in Universe.
Author Response
This is an interesting application of the ideas put forward in refs.[12,13], proposing a reduction of
the full Poincare symmetry within the Lorentz group to potentially mimick the effects of neutrino masses, to the computation of one-loop radiative effects in VSR QED. (The paper title is misleading since what is done in the paper is only VSR QED.)
I would suggest the following improvements before the paper is accepted.
1) Even though this is almost trivial, it may be beneficial to the less informed reader to show why the pole of the electron propagator subject to the nonlocal term in (1) is M_e=\sqrt{M^2+m^2} to lowest order in e.
I HAVE ADDED EQUATIONS. (2-3) TO EXPLAIN THIS POINT.
2) It cannot hurt to be more explicit about the Mandelstam-Leibbrandt regularization since in Eqs.(5),(6),(7),(9) the number \omega(=2-d/2) appears without any introduction.
I HAVE ADDED CHAPTER 3 TO EXPLAIN THE METHOD.
3) Fig.1 does not represent the entire vertex structure of the theorxy, rather a trunction of (n D)^{-1}
up to second order in A_\mu is presented. This should be mentioned and justified.
I INSERTED EQUATION (4) TO CLARIFY THIS POINT.
4) At several places a plural instead of a singular and other misprints appear:
differents -> different, Neanmoins -> Nevertheless, non -> none, preserve -> preserves, forms -> form, must holds -> must hold
I CORRECTED THIS.
5) In the third-last equation on p.6: \vec{s} is the spin vector (\hbar=1). How can this formula be obtained from the form-factor calculation?
Please explain.
I HAVE ADDED A PARAGRAPH THAT ENDS IN EQUATION (21) TO EXPLAIN THIS.
Upon satisfactory addressation of the above points the paper can be published in Universe.
I WANT TO THANK THE REFEREE FOR HIS VERY USEFUL COMMENTS THAT HAVE ENRICHED THE PRESENTATION OF THE PAPER.
Reviewer 2 Report
This is an interesting study of Very Special Relativity (VSR) as applied to the Standard Model (SM). The paper investigates the tree level Feynman rules, Ward identities and 1-loop corrections to the VSR version of QED. The paper is interesting and could be published but I have some questions/reservations.
First, the opening paragraph misspells "precission" (should be precision) and "Neanmoins" is French for "However" and in any case should have an accent mark over the "e".
Second, the author claims the mu-->e+ gamma is forbidden in the SM. It is only forbidden at tree level. A 1-loop calculation gives a non-zero (but at present undetectable small) decay rate for this process. (See the explicit calculation in "Gauge Theory of Elementary Particles" by Cheng and Li in section 13.3)
Third, the author says he is using the Feynman gauge (this is determined by the last term in eqn. 1). However isn't the Feynman gauge with a factor of (1/2) not (1/4)? See for example "Quantum Field Theory" 2nd ed. L. Ryder eqn. 4.76 and the surrounding discussion and also eqns, 7-11. 7.12. 7.13).
The author has checked that the results of the modified Feynman rules to preserve things like the anomalous magnetic moment of the electron, masslessness of the photon and bound on the electric dipole moment of the electron.
After the author addresses the above questions/comments the paper can be published.
Author Response
This is an interesting study of Very Special Relativity (VSR) as applied to the Standard Model (SM). The paper investigates the tree level Feynman rules, Ward identities and 1-loop corrections to the VSR version of QED. The paper is interesting and could be published but I have some questions/reservations.
First, the opening paragraph misspells "precission" (should be precision) and "Neanmoins" is French for "However" and in any case should have an accent mark over the "e".
I HAVE CORRECTED THIS.
Second, the author claims the mu-->e+ gamma is forbidden in the SM. It is only forbidden at tree level. A 1-loop calculation gives a non-zero (but at present undetectable small) decay rate for this process. (See the explicit calculation in "Gauge Theory of Elementary Particles" by Cheng and Li in section 13.3)
THE SM I AM REFERRING TO IS THE WEINBERG-SALAM MODEL. THERE NEUTRINOS ARE MASSLESS AND THE LEPTONIC QUANTUM NUMBER CORRESPONDING TO ELECTRON, MUON AND TAU ARE SEPARATELY CONSERVED. THIS IS TRUE AT ALL ORDERS IN THE LOOP EXPANSION, SO mu-->e+ gamma IS FORBIDDEN IN THE W-S MODEL AT ALL ORDERS IN PERTURBATION THEORY. THE PROCESS THAT IS DESCRIBED IN SECTION 13.3 OF CHENG AND LI NEEDS NEUTRINO MIXING WHICH IS NOT ALLOWED IN THE W-S MODEL, WHERE NEUTRINOS ARE MASSLESS. THIS STATEMENT IS ALSO IN CHENG AND LI 'S BOOK. THE PARAGRAPH BEFORE EQUATION (13.77) SAYS:''AS IT IS A LEPTON FLAVOUR-CHANGING PROCESS IT IS STRICTLY FORBIDDEN IN THE STANDARD THEORY WITH $M_\NU=0$ AND MUON NUMBER CONSERVATION.''
Third, the author says he is using the Feynman gauge (this is determined by the last term in eqn. 1). However isn't the Feynman gauge with a factor of (1/2) not (1/4)? See for example "Quantum Field Theory" 2nd ed. L. Ryder eqn. 4.76 and
the surrounding discussion and also eqns, 7-11. 7.12. 7.13).
I HAVE CORRECTED THIS.
The author has checked that the results of the modified Feynman rules to preserve things like the anomalous magnetic moment of the electron, masslessness of the photon and bound on the electric dipole moment of the electron.
YES.
After the author addresses the above questions/comments the paper can be published.
Submission Date 29 November 2018
Date of this review 14 Dec 2018 21:33:52
I WANT TO THANK THE REFEREE FOR HIS USEFUL REMARKS.